# More Than Meets the Eye: Revisiting the Roles of Heat Shock Factor 4 in Health and Diseases

**DOI:** 10.3390/biom11040523

**Published:** 2021-03-31

**Authors:** Saiful Effendi Syafruddin, Sheen Ling, Teck Yew Low, M Aiman Mohtar

**Affiliations:** UKM Medical Molecular Biology Institute, Universiti Kebangsaan Malaysia, Jalan Yaacob Latiff, Bandar Tun Razak, Kuala Lumpur 56000, Malaysia; ling_sheen@yahoo.com (S.L.); lowteckyew@ppukm.ukm.edu.my (T.Y.L.); m.aimanmohtar@ppukm.ukm.edu.my (M.A.M.)

**Keywords:** heat shock factors, cellular stress, cells development, transcription regulation, cataract, cancer

## Abstract

Cells encounter a myriad of endogenous and exogenous stresses that could perturb cellular physiological processes. Therefore, cells are equipped with several adaptive and stress-response machinery to overcome and survive these insults. One such machinery is the heat shock response (HSR) program that is governed by the heat shock factors (HSFs) family in response towards elevated temperature, free radicals, oxidants, and heavy metals. HSF4 is a member of this HSFs family that could exist in two predominant isoforms, either the transcriptional repressor HSFa or transcriptional activator HSF4b. HSF4 is constitutively active due to the lack of oligomerization negative regulator domain. HSF4 has been demonstrated to play roles in several physiological processes and not only limited to regulating the classical heat shock- or stress-responsive transcriptional programs. In this review, we will revisit and delineate the recent updates on HSF4 molecular properties. We also comprehensively discuss the roles of HSF4 in health and diseases, particularly in lens cell development, cataract formation, and cancer pathogenesis. Finally, we will posit the potential direction of HSF4 future research that could enhance our knowledge on HSF4 molecular networks as well as physiological and pathophysiological functions.

## 1. Introduction

Cells are constantly being exposed to various forms of cellular stresses and external insults. Therefore, it is vital for them to mount appropriate responses and adapt to these stresses in order to maintain proper cellular homeostasis and functions [1]. One of these responses include the activation of stress-associated responsive programs such as the heat shock response (HSR) [2], DNA damage response (DDR) [3], unfolded protein response (UPR) [4], hypoxic stress response [5], and oxidative stress response [6]. Since cells are dependent on adaptive response machinery to cope with these insults, dysregulations in any of these stress-associated responsive programs have been implicated in many diseases such as cardiovascular diseases, neurodegenerative diseases, inflammatory diseases, and cancer [7,8]. The heat shock response program, in particular, plays a critical role in facilitating cell adaptation and survival in elevated temperature environments and in other stressful conditions. The cellular heat shock response program is tightly governed by a family of transcription factors known as the heat shock factors (HSFs). This review focuses solely on heat shock factor 4 (HSF4), and we discuss the current understanding of the molecular properties and roles of HSF4 in health and diseases. Finally, we provide insights on the potential future HSF4 research direction to enhance our knowledge on HSF4 in general and its functional relevance in both physiological and pathophysiological processes.

## 2. Overview of the Heat Shock Response (HSR)

The HSR was initially characterized as a cellular program activated in response to an increase in temperature. It was later discovered that, in addition to elevated temperatures, other stimuli such as free radicals, oxidants, and heavy metals are able to trigger the HSR [9]. The heat shock proteins (Hsps) are downstream effectors of the HSR program. They act as molecular chaperones to facilitate proper protein folding as well as to prevent the accumulation of unfolded/misfolded proteins formed as a result of cellular perturbations or stressors [10]. Therefore, the stress response machinery serves to protect cells from proteotoxic stress and severe cellular damages by maintaining proteostasis and normal functions [10]. Like most cellular and molecular processes, the activation of HSR is tightly regulated by a specific transcription factor family known as heat shock factors (HSFs). Seven members of the HSF family have been identified in mammals to date, which are: HSF1, HSF2, HSF3, HSF4, HSF5, HSFX, and HSFY [11]. The human genome encodes all these HSFs except for HSF3, which is only found in birds, mice, and lizards [12,13,14]. Phylogenetic analysis of human HSFs has revealed that HSF1, HSF4, and HSF2 are evolutionarily closer to each other in the phylogenetic tree (Figure 1). Collectively, HSF1, HSF2, and HSF4 share several similar structural features (i) a highly conserved N-terminal winged helix-turn-helix DNA-binding domain (DBD), (ii) a leucine zipper oligomerization domain that comprises two heptad repeats HR-A/B (LZ1-3), (iii) a regulatory domain, and (iv) a C-terminus transactivation domain [15]. Besides, HSF1 and HSF2 contain an additional leucine zipper heptad repeat HR-C (LZ4) at the C-terminus that is absent in HSF4 [15,16] (Figure 2). This LZ4 domain represses HSFs oligomerization, which is required for their activation and binding to the target genes, by interacting with LZ 1–3 domain [17].

In response to thermal stress or other related stressors, HSFs undergo oligomerization to form a trimer that allows HSFs to bind to the heat shock element (HSE), made of inverted pentametric repeats “nGAAn”, located on the regulatory regions of their downstream target genes [18,19]. Since HSF5, HSFX, and HSFY lack HR-A/B oligomerization domain, it remains unclear whether they can function as transcriptional regulators. Out of the identified six human HSFs, HSF1, and HSF2 have been extensively studied and shown to be involved in various physiological processes as well as in diseases [20,21,22,23]. On the contrary, the functions of HSF5, HSFX, and HSFY remain little known. In addition to its role in spermatogenesis [24,25,26], HSF5 was found to be downregulated in lung adenocarcinoma patients within the Cancer Genome Atlas (TCGA) cohort. Its expression was also found to positively correlate with patients’ overall survival, inflammatory responses, and immune cell infiltration [27]. In contrast, HSFY has been reported to be involved in germ cell development, whereby the deletion of HSFY results in male infertility [28,29]. The molecular properties and roles of HSF4 in physiological and pathophysiological processes will be discussed in the subsequent sections.

## 3. Heat Shock Factor 4

### 3.1. Gene Structure

The human HSF4 gene was first isolated and characterized via cDNA library screening [16]. This gene is located on the long arm of chromosome 16, Chr16q22.1. The expression of HSF4 is generally ubiquitous across different tissue types; nonetheless, high expression of HSF4 is detected mainly in muscle tissues and also the cerebral cortex, midbrain, retina, and pancreas [30]. According to the Ensembl database (human assembly GRCh38.p13), the HSF4 gene can be transcribed into 25 different transcripts, albeit only eight transcripts are annotated as protein-coding [31] (Table 1). Comprised of 13 exons, the HSF4-215 and HSF4-225 are the two primary HSF4 isoforms, which are also annotated as HSF4b and HSF4a, respectively. These isoforms arise due to the alternative splicing that occurs at HSF4 exon eight and exon nine [32] (Figure 3a,b). HSF4b is the predominant HSF4 isoform (hereinafter referred to as HSF4 unless stated otherwise) that encodes 492 amino acids, whereas the HSF4a isoform is 30 amino acids shorter.

When HSF4 was first isolated, it was reported to function as a transcriptional repressor by negatively regulating the expression of Hsp27, Hsp70, and Hsp90 in HeLa cells. The discovery of two HSF4 isoforms by Tanabe et al. and subsequent independent studies revealed that the isoforms have opposing roles whereby HSF4a functions as a transcriptional repressor and HSF4b as an activator [32,33,34,35]. Interestingly, HSF4a has also been shown to repress HSF1 and HSF2-modulated transcriptional activities [34,35], indicating that there is crosstalk between HSF protein families in modulating the heat shock response programs.

### 3.2. Protein Structure

Structurally, HSF4 contains several important functional domains that endow them with the ability to function as a transcriptional regulator. First, HSF4 possesses a winged helix-turn-helix DNA-binding domain that binds to the heat shock elements of its target genes [36]. This domain, which is highly conserved across the human HSF family, is located at the N-terminus between amino acids 19 and 123. The next domain is the oligomerization domain that is essential for DNA binding and transcriptional regulatory activities of the HSF family members [37,38]. The HSFs monomer oligomerization is mediated by the intermolecular interaction of two hydrophobic leucine-zipper heptads repeat, HR-A and HR-B, that is located between amino acids 130–203 [37,38]. Unlike HSF1 and HSF2 that only oligomerize fully in response to stresses, HSF4, on the other hand, exists in an oligomeric form in its native state due to the absence of a third hydrophobic heptad repeat HR-C [16,32]. This HR-C domain negatively regulates the oligomerization process by interacting with the HR-A/B domain in order to prevent spontaneous or aberrant HSFs activities [17,37,39]. Since HSF4 lacks this HR-C domain, HSF4 is therefore constitutively active and has diverse functions that are not only limited to regulating the heat or stress response transcriptional programs.

Moreover, HSF4 bears two transactivation domains (TADs) that are located at the C-terminus [36]. These transactivation domains recruit and interact with relevant transcription factors, co-factors, chromatin-modifying factors as well as other components of the transcriptional apparatus to regulate the expression of its target genes. For instance, HSF4 transactivated the expression of HSPs during the cell cycle G1 phase by recruiting the Brahma-related gene 1 (BRG1), a component of the SWI/SNP chromatin remodeling complex, to the promoter region of the HSP-encoding genes [40]. HSF4 also contains a small regulatory domain (RD) that is flanked by these two TADs [41]. The function of this regulatory domain is still not fully understood, but it is postulated that this domain could be the site for post-translational modifications (PTMs) similar to those seen in HSF1 RD [36].

To date, there is only a portion of HSF4 crystal structure being successfully resolved, which is its DNA-binding domain. This crystal structure was generated using X-ray crystallography at 1.20 Å resolution. The Protein Data Bank (PDB) entries for these crystal structures are 6j6v and 6j6w, corresponding respectively to the wild-type and mutant HSF4 DNA-binding domain. To gain an insight into the complete HSF4 3D structure, we resorted to predicting it based on the entire amino acid sequence of HSF4 (UniProtKB ID: Q9ULV5) using I-TASSER (Iterative Threading ASSEmbly Refinement) [42,43,44]. As predicted by I-TASSER, the closest analog for HSF4 is the PSCD-region of the cell wall protein pleuralin-1 (derived from NMR) with a TM-score of 0.903, RMSD 1.00, sequence identity of 0.082, and coverage alignment of 0.917 (PDB ID: 2nbi) [45]. The TM-score depicts the topological similarity between the query structure and the known deposited PBD structure in the range of 0–1. A TM-score >0.5 correlates with the protein pairs of similar folds [42]. By superimposing these two structures using PyMOL (The PyMOL Molecular Graphics System, Version 2.0 Schrödinger, L.L.C), the PSCD C-terminal domain (PSCD5) of 2nbi model superimposed onto 6j6v, the wild-type HSF4 DNA-binding domain structure. In addition, I-TASSER predicted five different HSF4 structural models of which model three seemed to be the best fit because it superimposed perfectly onto the (i) 6j6v structure on its N-terminus, corresponding to the N-terminal of HSF4 being the DNA-binding domain, and the (ii) 2nbi structure (Figure 4).

### 3.3. Post-Translational Modifications

Phosphorylation, sumoylation, ubiquitylation, and acetylation are common post-translational modifications (PTMs) that are found in transcription factors, which can alter their regulatory activities. Upon inspecting the UniProt and PhosphositePlus^®^ databases, there were three phosphorylation sites (S298p, T471p, and S491p), two sumoylation sites (K287-sm and K293-sm), and one ubiquitylation site (K206-ub) reported for human HSF4 thus far (Figure 5). Importantly, HSF4b was demonstrated to possess a phosphorylation-dependent sumoylation motif, i.e., PDSM (ΨKxExxSP) at residues 293–300. The phosphorylation of the HSF4b serine 298 residue by mitogen-activated protein kinase (MAPK) ERK1/2 mediated the sumoylation of lysine at residue 293, which subsequently led to the suppression of HSF4b transcriptional activation function [46]. For instance, phosphorylation-mediated sumoylation of this HSF4b lysine residue reduced the expression of the crystallin gene, an important HSF4 downstream target that was involved in governing normal lens development and homeostasis [47]. On top of inhibiting HSF4b from activating the expression of its downstream targets, this specific modification could also promote HSF4b interaction with a transcriptional repressor to negatively regulate the expression of its target genes [48]. These findings highlighted the importance of HSF4b PTMs, which allow HSF4b to interact with different protein partners in a context-dependent manner. Such specific interaction is important in dictating HSF4b regulatory functions, whether acting as transcriptional activator or repressor.

Meanwhile, alanine-scanning assays revealed that the phosphorylation of HSF4b at threonine 471 residue by MEK6 mediated HSF4b localization into the nucleus [49]. The substitution of threonine to alanine at this residue caused HSF4b dissociation from the importin β-1/Hsc70 complex, which consequently inhibited its translocation into the nucleus and subsequent transcription regulatory activities [49]. Other than being the subject of phosphorylation by protein kinases, HSF4b could also be the target of dephosphorylation by ERK-specific dual-specificity phosphatase 26 (DUSP26), whereby the phosphorylation and dephosphorylation status of HSF4b would alter its transcriptional regulatory activities [50]. Furthermore, it was reported that BCAS2 interacted and negatively regulated HSF4 protein stability and transcription regulatory activity [51]. Mechanistically, it was demonstrated that BCAS2 mediated HSF4 proteasomal degradation via the ubiquitination of lysine 206 residue [51]. In line with this, either BCAS2 depletion or the substitution of lysine to arginine at this position (K206R) reduced HSF4 ubiquitination that in turn increased HSF4 protein stability [51].

## 4. The Roles of HSF4 in Normal Physiological Processes

As aforementioned, HSF4 is constitutively active due to the absence of a C-terminal HR-C domain that negatively regulates HSF4 oligomerization and activation [15,16]. Therefore, it has been suggested that HSF4 possesses diverse physiological functions that are not only limited to regulating the heat or stress-responsive transcriptional programs. Indeed, it has been demonstrated that HSF4 is involved in several physiological processes, including regulating cell differentiation and proliferation during the developmental process and modulating the DNA damage repair. Moreover, due to HSF4 roles in regulating these important cellular processes, HSF4 genetic alterations or aberrant activities have also been implicated in diseases such as cataracts and cancer.

### 4.1. Lens Development

#### 4.1.1. Lens Cell Proliferation and Differentiation

The tissue developmental process is a highly regulated process orchestrated by spatiotemporal expression and crosstalk between the tissue-specific transcription factors, co-factors, and chromatin remodeling components [52]. Lens are composed of two different cell types, which are the lens epithelial and lens fiber cells. Lens development is precisely coordinated that involves a tight regulation of lens epithelial cells proliferation and differentiation into fiber cells, expression of lens-associated structural proteins, denucleation, and organelle degradation. HSF4 has been widely demonstrated to play central roles in regulating lens development during eye organogenesis [53,54,55,56]. The expression of crystallins, major lens structural proteins, have been shown to be directly transactivated by HSF4 [53,54,55,56]. HSF4 knockout mice had abnormal lens development and the formation of cataracts due to aberrant cellular proliferation and differentiation, partly due to the reduced γ-crystallin expression in the lens fiber cells [53]. Besides, HSF4 also participated in the maturation and nucleus exportation of crystallins and HSP25 mRNA by interacting with RNA helicase UAP56 in the lens cells [57].

Moreover, p53 has also been reported to modulate lens cell proliferation and differentiation [58,59,60], in line with its roles as a key regulator of cell growth and death. HSF4 was reported to facilitate p53 stabilization and transcriptional activities during lens development that resulted in cell cycle arrest during the G1/S phase [61]. This, in turn, suppressed the lens epithelial cells proliferation and subsequently promoted their differentiation into the secondary fiber cells [61]. Besides, the stabilization of p53 by HSF4 also resulted in the upregulation of the p53-mediated apoptotic pathway that was essential in the lens fiber cells terminal differentiation [62]. Additionally, the expression of crystallin itself can also be transactivated by p53, which could then modulate the activities of its upstream regulators in driving normal lens development. This highlights the importance of the HSF4-p53-crystallin axis in regulating this process [60,62,63,64].

In addition to the HSF4-p53-crystallin axis discussed above, HSF4 also interacts with fibroblast growth factors (FGFs) in governing the lens development process [53,65]. HSF4 inhibition in mice resulted in the upregulation of FGF1, FGF4, and FGF7 expression, denoting HSF4 functions as FGFs transcriptional repressor [53]. HSF4 repressed the expression of these FGFs by competing with HSF1, a known FGFs activator, for a binding site at the FGFs heat shock element [53]. The expression of these FGFs needs to be tightly regulated because any imbalance or aberrant expression would lead to abnormal lens epithelial cell proliferation and differentiation [53,66]. However, unlike FGF1, FGF4, and FGF7, the expression of FGF2 did not get upregulated upon HSF4 inhibition in mice [53]. It was later demonstrated that instead of being transcriptionally regulated by HSF4, FGF2 has been found to be involved in promoting HSF4 protein stability and transcriptional activities by inducing the ERK1/2 pathway-mediated phosphorylation-dependent sumoylation in HSF4 [65]. These findings seem to suggest that the FGF2-HSF4 axis also plays critical roles in regulating the process of lens development, either by activating the expression of crystallin or repressing the expression of FGF1, FGF4, and FGF7 [65]. Besides, HSF4 has also been shown to modulate the expression of other lens structures and lens development-related genes such as beaded filament proteins (BFSP), vimentin, Src kinase-associated phosphoprotein 2 (SKAP2) [55,67,68]. The inhibition of HSF4 altered the expression of these genes, which subsequently impaired the lens cell differentiation and normal lens development.

#### 4.1.2. Lens Cell Homeostasis

Proteolytic machinery, like the lysosomal degradation pathway, plays essential roles in supporting lens epithelium proteostasis as well as the fiber cell terminal differentiation and nuclear degradation [69]. In line with this, the HSF4-crystallin axis is involved in maintaining the optimal level of lysosomal acidification in the lens cells by preventing the lysosomal transmembrane proton pump ATP6V1A from being degraded by the ubiquitin-mediated proteasomal degradation pathway [70,71]. Mechanistically, crystallin formed a protein complex with ATP6V1A and mTORC1, whereby the inhibition of HSF4, crystallin, or mTORC1 led to the dissociation of this complex and ATP6V1A destabilization, which consequently increased the lysosomal pH level [71]. Meanwhile, in line with the previously reported HSF4-BRG1 interaction in regulating HSPs expression [40], HSF4 was also found to recruit BRG1 to the DNase 2β (DLAD) regulatory elements and directly transactivated the expression of this gene during the lens fiber cell terminal differentiation [72,73]. DLAD functions were essential for the lens fiber cell denucleation and nuclear DNA degradation, whereby the accumulation of undegraded nuclear DNA contributed to abnormal lens development and the formation of cataracts [69,74].

#### 4.1.3. Lens Cell Protection and Survival

Another prominent role of HSF4 is that HSF4 could protect lens cells from oxidative stress, DNA damage, and drug-induced apoptosis during lens development. The lens cells are vulnerable to stress or damage due to a lack of protein turnover, in which the properties and functions of these cells have to be appropriately maintained throughout the lifetime [75]. To respond to these insults, the antioxidant heme oxygenase 1 (HMOX-1) plays a critical role in providing a defense mechanism against oxidative stress and the production of ROS in the lens cells [76,77]. In this regard, HSF4 was shown to transcriptionally activate the expression of HMOX-1 in lens epithelial cells by binding to the HSEs at the HMOX-1 promoter region [78]. DNA damage has been implicated in cataractogenesis, and HSF4 has been demonstrated to play a role in DNA damage repair by directly transactivating Rad51 expression, which is involved in homologous recombination (HR)-mediated DNA double-strand breaks [79]. Targeting HSF4 in zebrafish reduced the Rad51 expression and promoted the accumulation of DNA damage in the zebrafish lens [79]. Whilst ERK1/2 signaling pathway modulated DNA damage response and ERK1/2 has been shown to promote HSF4 transcriptional activity [50,80], it still remains unknown whether HSF4 regulated Rad51 expression in lens cells in ERK1/2 signaling pathway-dependent manner. In line with these findings, several reports have also associated HSF4 with anti-apoptotic activities in lens epithelial cells via its downstream effectors, crystallin and Hsp25 [70,81]. On the other hand, HSF4 has also been reported to protect lens cells from these insults by negatively regulating HSF1 expression and activities [81]. HSF1 has been shown to play a role in activating cell apoptosis in response to proteotoxic stress [82,83]. Mechanistically, in the lens cells, HSF4 promoted HSF1 degradation via lysosomal and proteasomal degradation pathways that consequently led to the inhibition of HSF1 transcriptional programs [81]. These studies further corroborated the importance of highly coordinated interplays between HSF4 and HSF1 during lens development and homeostasis [53,54,84].

### 4.2. Neuronal Genesis

Whilst the indispensable functions of HSF4 in regulating lens development have been widely studied, its role in the developmental process of other cells or tissues type still remains elusive. To date, there was only a report demonstrating HSF4 involvement in regulating the olfactory epithelium cells genesis in the nasal cavity. Similar to the observations in the lens, HSF4 also antagonized HSF1 functions in the olfactory epithelium cell genesis by competing for binding at the leukemia inhibitory factor (LIF) heat shock elements [85]. In this regard, HSF1 transcriptionally repressed the expression of LIF, whereas HSF4 opposingly transactivated the expression of this gene [85]. A normal level of LIF expression has to be maintained because altered expression of this gene resulted in impaired olfactory epithelium cell growth and the inhibition of olfactory sensory neuron maturation [86,87]. Consistent with these previous findings, HSF1-null mice had abnormal olfactory sensory neuron development, characterized by increased cell death, due to the upregulation of LIF expression [85]. Intriguingly, double HSF1 and HSF4 knockout mice had less severe olfactory development defects, potentially due to the partial restoration of LIF expression to the normal level [85].

An independent study by Homma et al. showed that loss of HSF1 in combination with either HSF2 or HSF4 aggravated the demyelination and astrogliosis of the murine central nervous system (CNS), highlighting the co-operation between HSF family members in maintaining CNS homeostasis [88]. Meanwhile, HSF4 was found to be co-expressed with DUSP26 in the mice brain cell lysates and in day 0 post-natal primary cortical neuronal cultures [50]. Whilst DUSP26 has been shown to be involved in neuronal differentiation [89,90,91], the roles of HSF4 in regulating this process and CNS development are yet to be functionally elucidated. Insights from studies in human lung cancer cells revealed that DUSP26 dephosphorylated HSF4, which consequently altered its DNA binding capability, by modulating the ERK1/2 signaling pathway activity [50]. These findings may suggest potential interactions between HSF4 and DUSP26, and the related signaling pathways in regulating CNS development, which will be interesting to investigate in future studies.

## 5. HSF4 Implications in Human Diseases

### 5.1. Cataracts

Cataracts, manifested by the clouding of the eye lens, is a progressive disease that commonly affects older people. This eye disease is highly prevalent, which accounts for almost half of the blindness cases worldwide. The age-related cataract is mainly due to the degradation of the proteins in the lens cells that are involved in supporting the lens structure and normal functions. In addition, cataracts can also be genetically inherited or caused by other comorbidities such as diabetes, injuries to the eye, and excessive exposure to ultraviolet light.

Since HSF4 is crucial in lens development, any dysregulations of HSF4 functions could hamper its function leading to a cataract. As discussed previously, knocking out HSF4 in mice resulted in abnormal lens development and the formation of early-onset cataract [53,55,56]. In line with these observations, mutations in the HSF4 gene have been widely associated with the pathogenesis of human early-onset or congenital cataracts. An early study in a Chinese congenital cataract patient cohort revealed that this disease was linked to hereditary mutations in the HSF4 DBD [92]. Mutation profiling in the HSF4 gene identified a T to C transition at the nucleotide 348 in all lamellar cataract-affected individuals that substituted the leucine to proline at the amino acid 115 [92]. In addition, the same group also analyzed a Danish family whose members were affected with congenital cataracts, as well as in sporadic cataract diseases, and found other missense mutations at the HSF4 DBD [92].

Interestingly, mutations at the HSF4 DBD were predominantly missense mutations that resulted in autosomal dominant disease [92,93,94,95,96,97]. Genetic alterations outside the DBD, such as those within or downstream the HSF4 HR-A/B oligomerization domain, have also been associated with congenital cataract [98,99,100,101]. In contrast, genetic alterations outside the HSF4 DBD were associated with autosomal recessive disease and composed of missense, frameshift, and nonsense mutations [92,93,94,95,99]. Functional and structural analyses on several of these reported mutations revealed that these mutations impaired the HSF4 DNA-binding and gene regulatory activity, which in turn would promote abnormal lens development and cataract formation [41,102,103]. There were five additional HSF4 mutations identified in a small fraction of age-related cataract patients [104]. Overall, the identification of these HSF4 genetic alterations and their associations with both congenital- and age-related cataracts further corroborated the HSF4 indispensable functions in governing normal lens development during eye organogenesis.

### 5.2. Cancer

Analysis of TCGA PanCancer genomic data revealed that HSF4 was rarely targeted by mutations in cancers [105,106]. Among the analyzed cancer types: bladder, uterine, esophagus, and stomach cancers had the highest frequency of HSF4 genetic alterations, which were about 3%–3.5% of all cases for each cancer type (Figure 6). With regards to the HSF4 expression level, this gene was found to be differentially expressed across different cancer types (Figure 7). This data was obtained by comparing the level of HSF4 expression in TCGA cancer sample cohorts with their respective normal tissues from both TCGA and GTEx cohorts (log2FC l2l; *p* < 0.05) [107]. HSF4 was significantly upregulated in clear cell renal cell carcinoma (ccRCC) as compared to the normal kidney tissues. In line with this, ccRCC patients who had high expression of HSF4 had a worse prognosis as compared to patients expressing a low level of HSF4 (Figure 8a,b). In contrast, the expression of HSF4 was significantly lower in glioblastoma multiforme, ovarian cancer, testicular germ cell cancer, thyroid cancer, and uterine carcinosarcoma as compared to their respective normal tissues (Figure 7). Further analysis revealed that there was no significant overall survival difference between patients who had high or low expression of HSF4 in glioblastoma, ovarian cancer, testicular germ cell cancer, and uterine carcinosarcoma (data not shown). Intriguingly, thyroid cancer patients who had high expression of HSF4 had a worse prognosis as compared to patients expressing a low level of HSF4 (Figure 8c,d).

Interestingly, independent analysis of the TCGA colorectal cancer and normal colon tissues RNA-Seq data by Yang et al. revealed that HSF4 was significantly upregulated in the colorectal cancer tissues [108]. Moreover, the survival analysis showed that patients with a high level of expression of HSF4 had poorer overall and recurrence-free survival than the low HSF4-expressing patients [108]. It is important to note that this analysis was performed using the UCSC Xena browser without the incorporation of the GTEx normal colon RNA-Seq data. A recent study showed that HSF4 expression was significantly upregulated in hepatocellular carcinoma (HCC) tissues, whereby targeting HSF4 reduced HCC cell growth and metastasis-associated phenotypes both in-vitro and in-vivo [109]. It was further demonstrated that HSF4 is involved in modulating the AKT-mTOR signaling pathway activation in a HIF1a-dependent manner [109]. In fact, deregulations of the PI3K-AKT-mTOR signaling pathway have been widely implicated in cancer pathogenesis due to their central roles in promoting cell growth and regulating cellular metabolism and macromolecule biosynthesis [110,111,112].

Whilst HSF4 transcriptionally activated HIF1a expression in HCC cells, HSF4 was reported to co-opt with HSF2 to repress HIF1a expression and subsequently inhibit the expression of HIF1a downstream target genes in breast and cervical cancer cells [113]. Intriguingly, overexpression of either HSF4 or HSF2 would displace each other from the complex and result in HIF1a transcriptional activation [113]. This finding demonstrated that the expression of HSF4 and HSF2 needs to be finely controlled for maintaining the homeostatic level of HIF1a expression. Therefore, dysregulated expression of either molecule would, in turn, activate the hypoxia-responsive transcriptional programs, which have been widely implicated in promoting tumorigenesis [114]. Moreover, HSF4 deletion in either p53 or Arf-deficient mice induced cellular senescence that in turn impaired cell growth and suppressed the formation of spontaneous tumor in these mice models [115]. The induction of cellular senescence was due to the upregulation of cyclin-dependent kinase inhibitors p21, in which these observations were similar to those observed in the HSF4-deficient mouse fibroblasts [115]. Concordantly, Hsp72 deletion inhibited Her2-induced mammary transformation due to the upregulation of p21 and subsequent induction of senescence-mediated cell growth inhibition [116,117].

## 6. Future Perspectives and Concluding Remarks

To date, the functions of HSF4 in governing lens development and homeostasis have been comprehensively elucidated. These insights include the HSF4 downstream transcriptional networks, upstream regulators, and post-translational modifications, which all play important roles in promoting normal lens cell proliferation and differentiation during eye organogenesis. Because these processes are finely tuned and centered around HSF4, normal HSF4 expression and functions need to be precisely maintained throughout the lens development process. Therefore, any slight adjustments to HSF4 expression would impair the lens cell proliferation and differentiation that would result in the formation of cataracts. In fact, a myriad number of studies have reported that mutations in HSF4 were implicated in the formation of congenital or age-related cataracts.

To understand the which HSF4 mutations that would instigate diseases, genome editing technology can be employed. For example, the emerging CRISPR-based genome editing tool can be used to systematically mutate the HSF4 gene to prioritize the dominant mutations leading to cataracts. The mutations responsible for cataracts could then be reversed using a similar approach. For example, the CRISPR-Cas9 gene-editing tool is currently being utilized as a potential approach for restoring the inherited genetic mutations in the CEP290 gene that causes the blindness disorder Leber’s congenital amaurosis 10 (LCA10) [118]. Alternatively, HSF4 mutants that lead to defective gene function can be restored by using CRISPR-activation (CRISPRa) approach. Such a technique has been successfully carried out in treating other inherited blindness [119]. The study showed that transactivation of photoreceptor-specific M-opsin (Opn1mw) resulted in prolonged retinal function and delayed retinal degradation in a mouse model. Hence, a similar approach could be used in the case of HSF4 to reverse the defective mutations or to activate HSF4 expression to prevent cataracts.

HSFs need to be in their trimeric state to exert their DNA binding and transcriptional regulatory activities. However, unlike HSF1 and HSF2, HSF4 is constitutively active due to the absence of the HR-C domain that negatively regulates this oligomerization process. Therefore, HSF4 is deemed to have diverse functions and not only limited to modulate the heat shock or stress-responsive gene expression programs. Indeed, as comprehensively discussed in the previous section, HSF4 functions are indispensable during lens development and olfactory sensory neuronal genesis. In fact, most insights on HSF4 roles in cellular physiological processes are in the context of regulating the lens cell differentiation, growth, and survival. According to the protein atlas database, in addition to retinal tissue, HSF4 is also significantly expressed in muscle, brain, and pancreas tissues. These observations are in line with Nakai et al. seminal report on the first human HSF4 gene isolation and characterization [16]. High expression of HSF4 may suggest that this gene could have relevant functions in these tissue types, similar to those observed in the lens cells. HSF4 was also found to interact with various determinants of cell growth and survival in the lens cells including p53, and proteins that are involved in MAPK and mTOR signaling pathways. Even though the tissue type-specific gene expression programs and interactome may influence HSF4 activity and functions, it is definitely worth interrogating HSF4 functions in the muscle, brain, or pancreas tissues, for instance. This is because the evidence highlighted above may support the notion of HSF4 potential roles in regulating these tissues’ physiological processes.

Whilst the association between HSF4 mutations and cataract formation has been comprehensively elucidated, there are also emerging reports implicating the association between HSF4 and cancer pathogenesis. Since cancer is a disease manifested by uncontrolled cell growth and cell death evasion [120], HSF4 involvement in supporting tumorigenesis could substantiate its potential roles in regulating growth and survival of diverse cells or tissue types and not only restricted to the lens. Meanwhile, HSF4 upregulation was found to be associated with poor prognosis in colorectal, liver, and kidney cancers, whereby targeting HSF4 in liver cancer cells impaired the cell growth both in-vitro and in-vivo. These findings were important and may highlight HSF4 functional relevance in supporting tumorigenesis. Since there have been only a couple of studies on HSF4 and cancer, more comprehensive investigations on HSF4 functions and their associated interactome in promoting cancer formation and progression are warranted in the future. This knowledge is vital in improving our current understanding of the enigmatic tumorigenesis process and would perhaps pave the way for the development of novel cancer diagnostic and/or therapeutic strategies.

On a separate note, there might be specialized roles of HSF4 in the clear cell renal cell carcinoma, which is the most common subtype of kidney cancer. According to the transcriptomics data, HSF4 expression was significantly upregulated in TCGA ccRCC clinical samples as compared to normal kidney tissues, and patients who had higher expression of HSF4 had worse overall survival [105,106]. VHL biallelic inactivation and the resultant pro-oncogenic HIFa accumulation contribute to 90% of all sporadic ccRCC cases, making the VHL-HIFa axis the hallmark gatekeeper of kidney tumorigenesis. Since HSF4 was found to transcriptionally regulate HIFa expression in liver cancer, it is thenceforth pertinent to determine whether HSF4 is also involved in controlling HIFa expression in ccRCC. If HSF4 is indeed regulating HIFa expression, HSF4 could then be among the important drivers of ccRCC pathogenesis, which is in line with its significant upregulation in ccRCC. Moreover, HSF4 modulated the AKT-mTOR signaling pathway activity in liver cancer, whereby this signaling pathway is among the key ccRCC clinical targets due to the frequent mTOR signaling pathway hyper-activation in ccRCC. On top of elucidating the possible interaction between HSF4 and HIFa, it is also noteworthy to interrogate the potential roles of HSF4 in modulating the mTOR signaling pathway in ccRCC. Collectively, this knowledge could improve the understanding of the ccRCC molecular landscape and pave the way for the development of novel ccRCC diagnostic and/or therapeutic strategies, potentially revolve around HSF4.

## Figures and Tables

**Figure 1 biomolecules-11-00523-f001:**
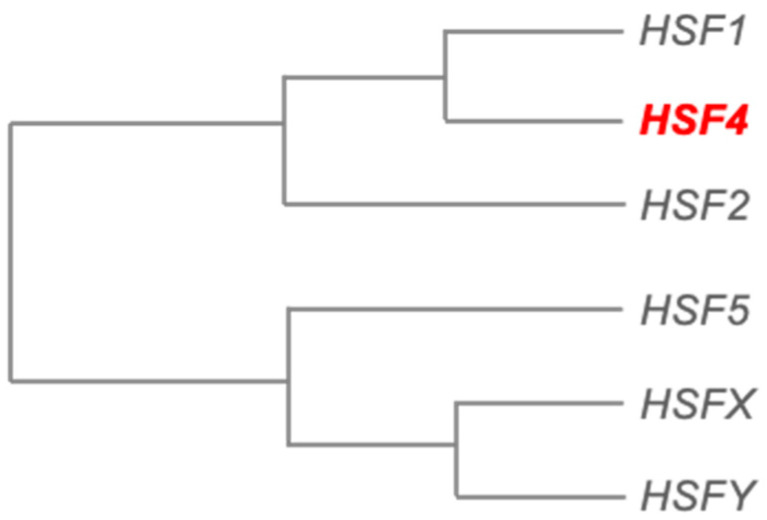
Phylogenetic tree of human heat shock factor (HSF) family members generated using MEGA-X tool (https://www.megasoftware.net, accessed on 15 December 2020). HSF4 is the closest to HSF1 in the phylogeny.

**Figure 2 biomolecules-11-00523-f002:**
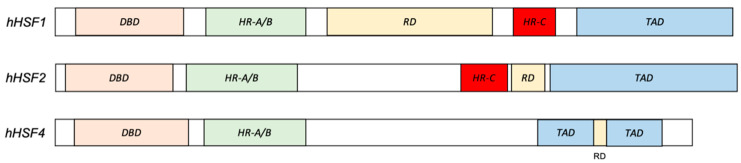
Comparison of human HSF1, HSF2, and HSF4 protein structure. All human HSF members possess a highly conserved DNA binding domain (DBD), oligomerization domain (HR-A/B), regulatory domain (RD), and the transactivation domain (TAD). In contrast to HSF1 and HSF2, HSF4 lacks the oligomerization repressor domain (HR-C).

**Figure 3 biomolecules-11-00523-f003:**
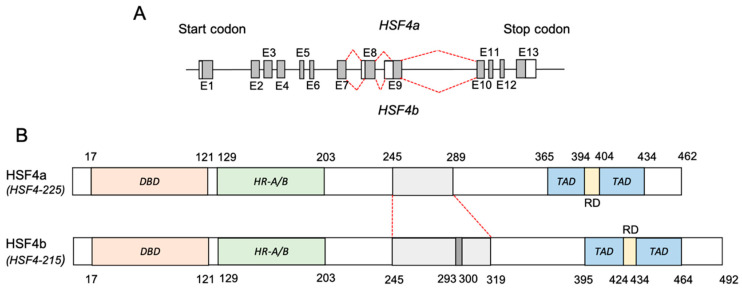
HSF4 gene structure and post-transcriptional modifications. (**A**) The HSF4 gene is comprised of 13 exons, which via alternative splicing within exon eight and nine can give rise to two major HSF4 isoforms: HSF4a (HSF4-225) and HSF4b (HSF4-215). (**B**) HSF4b, the predominant HSF4 transcript, is translated to produce 492 amino acids, 30 amino acids longer than its spliced variant counterpart. Whilst HSF4b functions as a transcriptional activator, HSF4a is reported to act as a transcriptional repressor. DBD: DNA binding domain; HR-A/B: oligomerization domain; RD: regulatory domain; TAD: transactivation domain.

**Figure 4 biomolecules-11-00523-f004:**
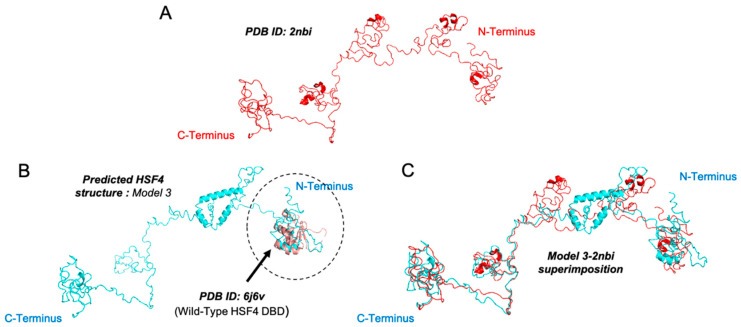
HSF4 3D-structure as predicted by I-TASSER. (**A**) I-TASSER predicted 2nbi as the closest HSF4 analog. The superimposition of the predicted HSF4 structure model 3 with the (**B**) previously resolved wild-type HSF4 DNA binding domain structure (Protein Databank (PDB) ID: 6j6v) and (**C**) its closest analog 2nbi.

**Figure 5 biomolecules-11-00523-f005:**
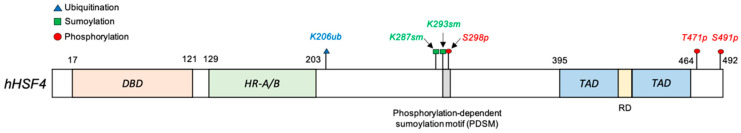
HSF4 protein post-translational modifications (PTMs). Three phosphorylation and two sumoylation sites and one ubiquitylation site have been reported for HSF4.

**Figure 6 biomolecules-11-00523-f006:**
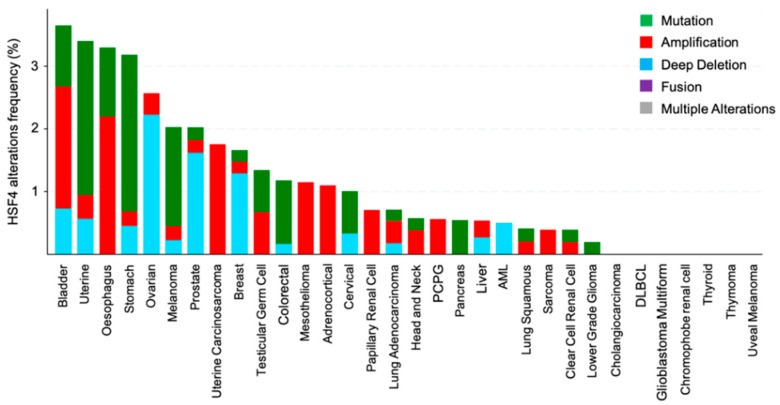
The frequency of HSF4 genetic alterations in thirty-two different cancer types characterized in the large scale Cancer Genome Atlas project. The identified genetic alterations in HSF4 include mutation, amplification, deep deletion, and fusion. These HSF4 genetic alterations frequency data were queried and extracted using cBioPortal (https://www.cbioportal.org/, accessed on 9 January 2021).

**Figure 7 biomolecules-11-00523-f007:**
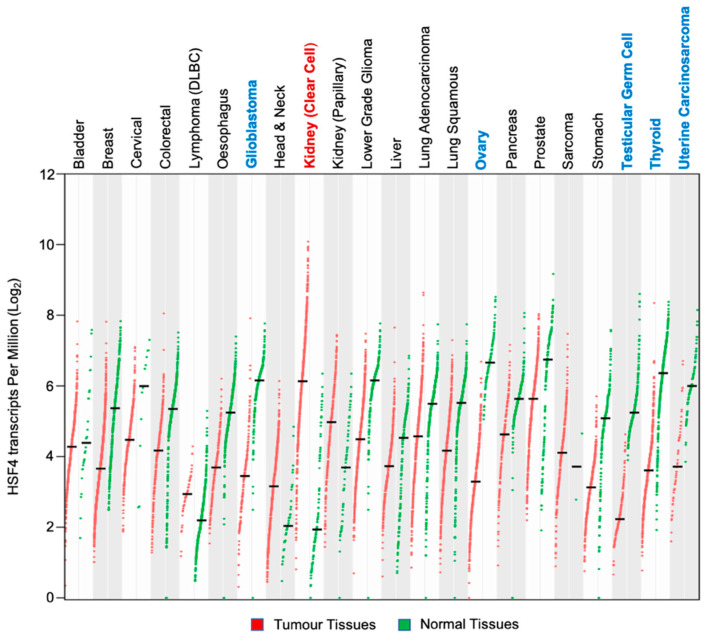
Dot plots representing the HSF4 expression RNA-Seq data in tumor and normal tissue samples from The Cancer Genome Atlas (TCGA) and The Genotype-Tissue Expression (GTEx) databases, respectively. HSF4 was found to be significantly upregulated in clear cell renal cell carcinoma (ccRCC) and significantly downregulated in glioblastoma, ovarian cancer, testicular germ cell cancer, thyroid cancer and uterine carcinosarcoma (Log2FC|2|; *p*-value < 0.05). The data were analyzed and visualized using the GEPIA2 (Gene Expression Profiling Interactive Analysis) web application (http://gepia.cancer-pku.cn, accessed on 9 January 2021).

**Figure 8 biomolecules-11-00523-f008:**
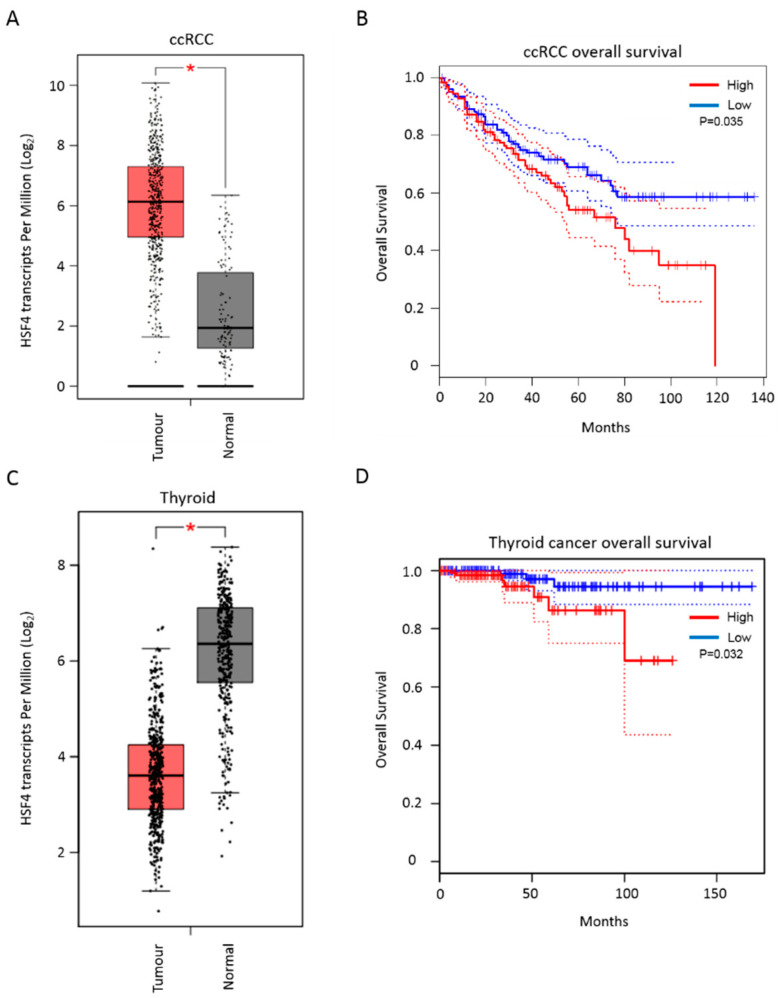
(**A**) Significant upregulation of HSF4 in TCGA ccRCC samples as compared to normal kidney tissues. (**B**) High expression of HSF4 was significantly associated with poor overall survival in TCGA ccRCC patient cohort. (**C**) Significant downregulation of HSF4 in TCGA thyroid cancer samples as compared to normal thyroid tissues. (**D**) High expression of HSF4 was significantly associated with poor overall survival in TCGA thyroid cancer patient cohort. Asterisk (*) indicates that there is statistically significant difference (log2FC l2l; *p* < 0.05) in the HSF4 expression between the tumour and normal tissues.

**Table 1 biomolecules-11-00523-t001:** List of annotated HSF4 protein-coding transcripts. The data were retrieved from Ensembl data using human assembly GRCh38.p13.

Name	Transcript ID	Length(bp)	Protein(aa)	Consensus Coding Sequence (ccds)	RefSeq Match
HSF4-215	ENST00000521374.6	1702	492	CCDS42175	NM_001374675.1
HSF4-225	ENST00000584272.5	1549	462	CCDS45510	NM_001374674.1
HSF4-203	ENST00000517685.5	1249	416		
HSF4-209	ENST00000519601.5	669	223		
HSF4-211	ENST00000520304.5	565	141		
HSF4-204	ENST00000517729.5	556	185		
HSF4-205	ENST00000517750.5	546	182		
HSF4-217	ENST00000521916.1	404	63

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
