# Peer review of "More Than Meets the Eye: Revisiting the Roles of Heat Shock Factor 4 in Health and Diseases"

_biomolecules, 2021, doi:10.3390/biom11040523_

Round 1

Reviewer 1 Report

I appreciate the very well-done work of authors who don't seem to have their own articles on this subject. The authors have performed several bioinformatics analyses, however, the review is based entirely on papers published by others. Nonetheless, it is comprehensive and logically organized. It was possible to include almost all relevant papers because the literature on the subject is not very extensive. The authors discuss the roles of HSF4 in health and diseases, especially in lens cell development, cataract formation, and cancer pathogenesis. The other possible functions of HSF4 are not well documented in the literature and are therefore not discussed here. I have a different opinion on the future perspectives, but the authors are entitled to their views.

I only have a few minor comments/suggestions/questions.

  1. It would be reasonable to include information on the regulation of HSF4 gene expression, if available. For example, it was found that: “The transcription activity of heat shock factor 4b is regulated by FGF2” (https://pubmed.ncbi.nlm.nih.gov/23200779/). As HSF4 acts as a transcriptional repressor of fibroblast growth factor (FGFs), it would be interesting to discuss possible relationships.
  2. Table 1: What is RefSeq Match for HSF4-225? It is not so important, however, the NCBI database shows different HSF4 transcripts (13 or 15 exons). Please check and decide.
  3. Figure 3: some topological corrections should be done (E10, E12, E13, RD).
  4. Figure 3 legend: please check if correct, especially in (B). DBD, TAD, and so on should be explained in the legend. Some punctuation mistakes.
  5. Line 115: “HSF4a function” or “HSF4a functions”?
  6. Lines 116-117: “One notable structural feature difference between these two isoforms is that the HSF4a lacks the transcriptional activation domain”. TAD domain is shown in Figure 3 in both isoforms. Do you mean HR-C?
  7. Lines 131-132: “Unlike HSF1 and HSF2 that only oligomerize in response to stresses”. HSF2 is believed to be a dimer and under the stress forms homotrimer or heterotrimer with HSF1. So it would be safe to write: “Unlike HSF1 and HSF2 that only oligomerize fully in response to stresses”.
  8. Lines 149-166: I don't understand what is going to come out of this part.
  9. Figure 5 and 6 legends: please remove unnecessary repetitions (like “for HSF4”).
  10. Line 207: should be “206 residue”. Lack of reference for described BCAS2 action: BCAS2 interacts with HSF4 and negatively regulates its protein stability via ubiquitination (https://pubmed.ncbi.nlm.nih.gov/26319152/)
  11. Line 240: “promote” or “promoted”?
  12. Line 271: “expression40” or “expression [40]”?
  13. Line 331: “commonly affecting” or “is commonly affecting”?
  14. Lines 332-334: “The age-related cataract is mainly due to the degradation of the proteins in the lens cells that is involved in supporting the lens structure and normal functions.” “that is” or “that are”?
  15. Line 346: “whose members affected” or “whose members were affected”?
  16. Figure 6: should be “HSF4 alterations frequency”, not HAF4. Since there is no - sign, + sign is not necessary here (relevant information in the legend should suffice). Please adjust the color legend (other shades of blue).
  17. Lines 372-375: “the expression of HSF4 was significantly lower in glioblastoma multiforme, ovarian cancer, testicular germ cell cancer, thyroid cancer, and uterine carcinosarcoma as compared to their respective normal tissues”. It would be reasonable to provide here information on how this relates to the survival of cancer patients.
  18. Figure 8: punctation.

Author Response

Thank you for the time to comprehensively review our manuscript. We truly appreciate the very positive overall feedback and comments/suggestions given. Please see the attachment for point-by-point responses

Reviewer 2 Report

In my opinion, the manuscript of Syafruddin et al. is an excellent, timely review paper focusing on the structure and functions of HSF4. The manuscript is well written and well organized. The Authors provide a summary of the molecular properties of HSF4 as well as its role under normal and pathological conditions. Finally the Authors outline the possibilities of therapeutic application of the accumulated knowledge. I have only two minor remarks regarding the manuscript:

The sentence in line 289 would probably sound correct as follows: Targeting HSF4 in zebrafish reduced the Rad51 expression and promote accumulation of DNA damage in the zebrafish lens [77].

The sentence “it was further demonstrated that the HSF4 also exerted its protective functions by negatively regulating HSF1 expression and activities [71]” in line 296 seems to be contradictory and may require further explanation.

Author Response

Thank you for the time to comprehensively review our manuscript. We truly appreciate the very positive overall feedback and comments/suggestions given. Please see the attachment for point-by-point responses

This manuscript is a resubmission of an earlier submission. The following is a list of the peer review reports and author responses from that submission.